# Comparison of adult versus elderly patients with abdominal trauma: A retrospective database analysis

Jeremy Dash[1]*, Elisabeth Andereggen[1], Imad Bentellis[2], Damien Massalou[3]

1 Departement of Digestive Surgery, Geneva University Hospitals, Geneva, Switzerland, 2 Departement of Urology, University Hospital of Nice, Nice, France, 3 Emergency Surgery Unit, Digestive Surgery, University Hospital of Nice, Nice, France

* dashjer0@gmail.com

**Data Availability Statement:** We fully support the principles of transparency and data sharing in scientific research. However, the data used in our

## Abstract

### Background

The growing geriatric population has specific medical characteristics that should be taken into account especially in trauma setting. There is little evidence on management of abdominal trauma in the elderly and this article compares the management and outcomes of younger and older patients in order to highlight fields of improvement.

### Method

We conducted a retrospective database analysis from two European university hospitals selecting patients admitted for abdominal injury and extracted the following data: epidemiological data, mechanisms of the trauma, vital signs, blood tests, injuries, applied treatments, trauma scores and outcomes. We compared to different age group (16–64 and 65+ years old) using uni- and multivariable analysis.

### Results

1181 patients were included for statistical analysis. The main mechanisms of injury in both group were traffic accidents and in the elderly group, falls were more frequent. Both had similar Abbreviated Injury Score except for the thoracic injuries, which was higher in the elderly group. We reported a death rate of 13% in the elderly group and 7% in the younger group. However, multivariable analysis did not report age as an independent predictor of mortality. The management including surgery, blood transfusion and need for intensive care were similar in both groups.

### Conclusion

Although elderly patients suffering abdominal trauma have an almost two fold higher mortality, their management is quite similar leading to an important point of improvement in regards to triage and lower threshold for more aggressive management and surveillance.

study includes sensitive information such as patient age, mechanism of trauma, year of the trauma, injuries and the length of stay in two recognizable hospitals. Due to the nature of these data points, there is a risk that individuals could be identifiable, thereby compromising patient confidentiality and privacy. Our research adheres to stringent ethical guidelines and complies with the privacy regulations set forth by our institutions and the New Federal Act on Data Protection in Switzerland and the General Data Protection Regulation in France. These regulations impose strict limitations on the sharing of personal health information to protect patient privacy. While we are unable to share the raw data publicly, we are committed to ensuring the transparency and reproducibility of our research. To this end, we can provide the contact information to which data requests may be sent : For the Geneva University Hospital registry : Dr Axel Gamulin Hôpitaux Universitaires de Genève Rue Gabrielle-Perret-Gentil 4 1211 Genève axel.gamulin@hcuge.ch 0041223723311 Commission cantonale d'éthique de la recherche CCER Rue Adrien-Lachenal 8 1207 Genève 0041 22 546 51 01 ccer@etat.ge.ch www.ge.ch/lc/ccer For the Nice registry (REGISTRY CIL n°272): Dr Damien MASSALOU Centre Hospitalier Universitaire de Nice Voie Romaine 30 CS 51069 - 06001 Nice Cedex 1 massalou.d@chu-nice.fr 0033663269264 / 0033492038614 Non-author point of contact: Commission nationale de l'informatique et des libertés 3 Place de Fontenoy TSA 80715 75334 PARIS CEDEX 07 0033153732222.

**Funding:** The author(s) received no specific funding for this work.

**Competing interests:** The authors have declared that no competing interests exist.

Age itself does not seem to be a reliable predictor of mortality. Introducing a frailty score when taking care of elderly trauma patients could improve the outcomes.

## Background

Improving knowledge and care for elderly trauma patients is particularly relevant in the context of an aging population. In Switzerland, the median age increased by more than 10 years from 1971 to 2019 and life expectancy at birth rose from 72.4 to 81.9 years [1]. This is a worldwide trend with the number of people over 65 projected to grow from an estimated 524 million in 2010 to nearly 1.5 billion in 2050, with the highest increase in developing countries [2].

The geriatric population represents 23% of all emergency department admissions for trauma, which is the fifth cause of death in this group of patients [3]. The most common mechanisms are falls, motor vehicle accidents and burns [4]. Patients over 65 years of age are twice more likely to die from trauma than younger patients with a similar injury severity score (ISS) [5]. Trauma in the elderly consists mainly of injuries to the head and the musculoskeletal system [6]. However, around 10% of seriously injured patients admitted to emergency departments with abdominal trauma are aged 65 and over. In this setting, surgeons, emergency and intensive care physicians will see a growing number of elderly patients suffering from trauma-related injuries requiring advanced and tailored management.

Taking care of elderly trauma patients is challenging. The diminished physiological response capacities and pre-existing comorbidities make this population particularly vulnerable. The management of these patients can pose diagnostic difficulties, for instance impaired physiological response and medications such as beta-blockers and steroids can hide the expected shock response leading to under-triage [7]. Due to this, the National Expert Panel on Field Triage of the American College of Surgeons recommends that patients over 65 with systolic blood pressure under 110 mmHg should be considered in shock and thus be transported to a trauma center (for younger patients the threshold is 90 mmHg) [8]. Trauma in the elderly presents therapeutic difficulties as well, especially if there is an association between lesions with a high risk of bleeding and the use of anticoagulation medication.

Until recently, non-operative management of abdominal trauma in elderly patients was considered too risky. Godley et al. concluded in 1996 that non-operative management of blunt splenic injury is contraindicated in patients over 55 years old [9]. However, more recent studies and guidelines have demonstrated that advanced age is not considered a contraindication to non-operative management in blunt splenic and hepatic injuries [10, 11].

There are few studies that specifically focus on abdominal trauma in the elderly population. Increased understanding of the characteristics of this population and the effectiveness of treatments administered can help to define management recommendations and identify areas for further research.

## Objective

The objective of this study is to compare the severity of abdominal trauma, management approaches, as well as associated morbidity and mortality between two groups of patients—those aged 65 years and older and a control group of patients aged 16–64 years. The aim is to identify factors that increase morbidity and mortality and determine whether older patients with abdominal trauma should receive different management.

## Method

This is a retrospective study, focusing on the analysis of epidemiological and clinical data from the "Severely Injured" registers of two university hospitals in Europe. The collected data included epidemiological data, circumstances and mechanisms of the trauma, vital signs, blood tests, injuries following trauma, applied treatments, trauma scores and outcomes.

The Cantonal Research Ethics Commission evaluated and accepted the use of the database for this study for one of the participating hospital and for the other the database is registered with the national committee for databases allowing the use of anonymized data.

We selected patients in the registers from 2010 and 2014 to 2019 because of lacking data in the register prior to these years. Inclusion criteria were admission to the emergency department (ED) for trauma with an abdominal AIS >0 (Abbreviated Injury Score). Patients selection's flowcharts are available on Figs 1 and 2. The data from both registers were collected between the 03/02/2020 and 13/03/2020.

The anonymized patients data included in the study were extracted from the registers and transferred to a computerized support (Excel), from which they have been processed and analyzed. The investigators did not have access to information that could identify individual participants during or after data collection.

For each patient, we extracted the following data: age, sex, mechanism of trauma, heart rate (HR), systolic and diastolic blood pressure (SBP, DBP), Glasgow Coma Scale (GCS) on admission in the ED, haemoglobin (hb), pH, arterial lactates, abdominal, thoracic, kidney, skeletal and head Abbreviated Injury Scale (AIS) scores, Injury Severy Score (ISS), surgical intervention, angio-embolization, transfusion, complications (Clavien Dindo classification), intensive care and hospital length of stay.

The threshold of 65 years of age for our definition of elderly or geriatric patient was selected arbitrarily, knowing the wide range of definitions used in literature (55–80 years old). The age of the control group (16–64) was used to enroll patients with similar physiological status corresponding to adults.

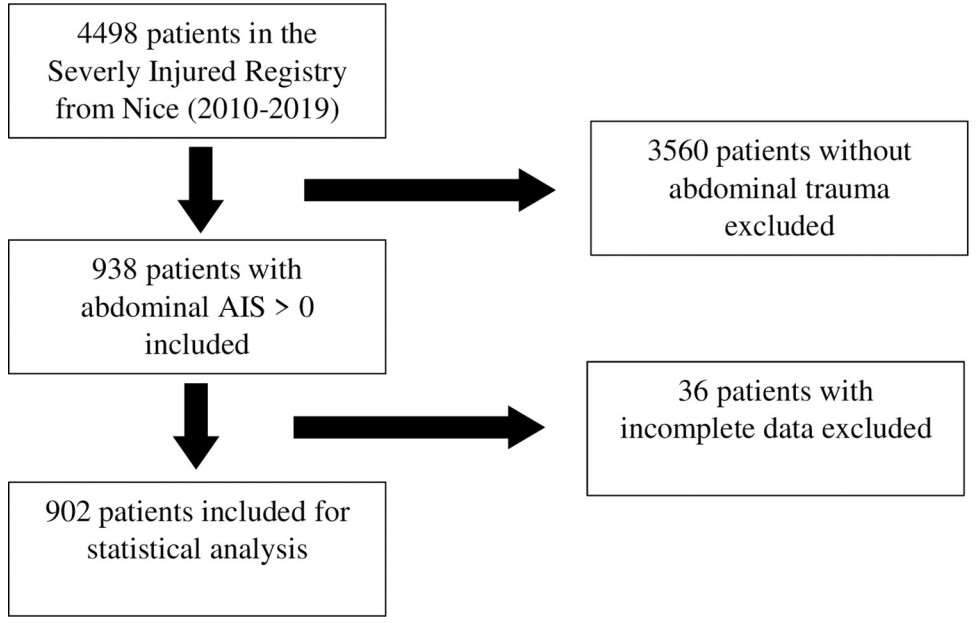

**Fig 1. Nice patients' selection flowchart.**

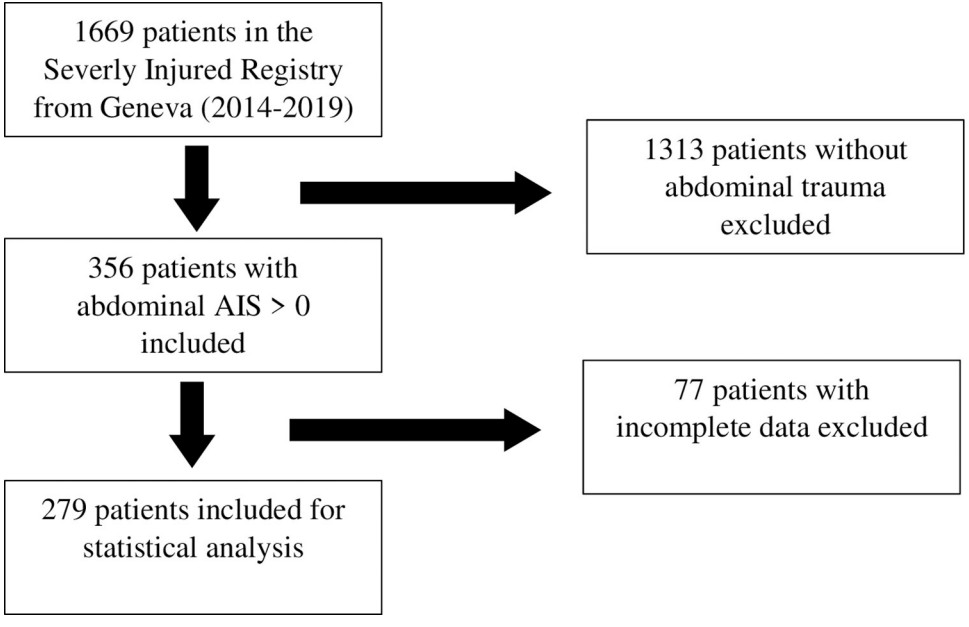

**Fig 2. Geneva patients' selection flowchart.**

The STROBE guidelines for case-control studies were used to ensure proper reporting of methods, results, and discussion.

Comparison between the two groups, 0–64 years old (young group (YG)) and 65 years and older (elderly group (EG)) was performed with the use of the mean with standard deviation to report continuous parametric data with the Student t-test to report the differences between the groups. We reported categorical data as numbers and percentages with the $\chi^2$ test to assess the difference. The Mann-Whitney U-test was used for ordinal score comparison. In all cases, we considered a $p$-value of <0.05 as statistically significant.

To assess predictive factors of operative management and mortality, we performed a logistic regression. The multivariable analyses was adjusted to the following factors (age, penetrating injuries, shock, GCS, SBP, DBP, hb, arterial lactates, abdominal and thoracic AIS and ISS). We used the R statistical software to perform statistical analysis.

Missing values (21 patients had no reported BP, 86 no reported arterial lactates, 53 no GCS and 23 no hb) probably due to lack of transcription in the patient's file, were considered to be missing completely at random and were omitted in the analysis without excluding other variables from the same patient using complete-case analysis. Patients with grossly lacking values were not included in our study.

## Results

6167 patients in the Severly Injured registers were reviewed. 4873 were excluded for not having abdominal trauma and 113 patient with non-interpretable data (grossly lacking values) were not included as well. In total, 1181 patients were included for statistical analysis: 279 from the first hospital and 902 from the second. The mean age was 39.6 and the male to female ratio was 3.26. 119 (10.1%) patients were 65 years old and older. The main mechanisms of injury were traffic accidents (55%), stab wounds (14.6%) and falls from heights (14.1%). Falls (14% vs 2%) and falls from heights (24% vs 13%) were predominantly in the EG and there were fewer stab wounds (7% vs 15%) and sport injuries (2% vs 4%).

**Table 1. Patients' characteristics.**

| Characteristics | All patients (n = 1181) | Elderly group (n = 119) | Young group (n = 1062) | p-value |
|---|---|---|---|---|
| Female n (%) | 277 (23%) | 45 (38%) | 232 (22%) | <0.001 |
| Age (mean+/-SD) | 39.5 +/-17.6 | 74.7 +/- 7.3 | 35.6 +/- 13.6 | |
| Traffic accident n (%) | 650 (55%) | 52 (44%) | 598 (56%) | **0.009** |
| Fall n (%) | 34 (3%) | 17 (14%) | 17 (2%) | <0.001 |
| Fall from heights n (%) | 166 (14%) | 29 (24%) | 137 (13%) | **0.001** |
| Stab wound n (%) | 172 (15%) | 8 (7%) | 164 (15%) | **0.01** |
| Gun shot wound n (%) | 51 (4%) | 5 (4%) | 46 (4%) | 0.947 |
| Sport accident n (%) | 47 (4%) | 2 (2%) | 45 (4%) | 0.176 |
| Other n (%) | 31 (3%) | 4 (3%) | 27 (3%) | 0.596 |
| Non specified (%) | 30 (3%) | 2 (2%) | 28 (3%) | 0.530 |
| Abdomen AIS (mean) | 3.15 | 3.01 | 3.16 | 0.133 |
| Thorax AIS (mean) | 1.50 | 1.84 | 1.46 | **0.008** |
| Kidney AAST (mean) | 0.45 | 0.10 | 0.49 | <0.001 |
| Skeletal AIS (mean) | 1.01 | 0.90 | 1.03 | 0.311 |
| Head AIS (mean) | 0.54 | 0.61 | 0.53 | 0.473 |
| ISS (mean) | 19.99 | 19.99 | 19.99 | 0.999 |
| Associated injury n (%) | 892 (76%) | 94 (79%) | 798 (75%) | 0.354 |

n: number, SD: standard deviation), significant p-values in bold.

Regarding the injury score, the mean abdomen, skeletal and head AIS and ISS were similar in both groups. The mean thorax AIS was higher in the EG (1.84 vs 1.46, p-value 0.008) and the kidney AAST score was lower (0.10 vs 0.49, p-value 0.0003) (details on Table 1).

On admission in the emergency department, elderly patients had a higher SBP (122.6 vs 112.9 mmHg, p-value 0.037), arterial pH (7.34 vs 7.28, p-value 0.037) and lower hb concentration 118.9 vs 126.7 g/l, p-value 0.004). DBP, HR, GCS and arterial lactates were similar in both groups with no significant statistical difference (details on Table 2).

There were 473 patients with spleen injuries, comprising 59 in the EG and 414 in the YG, accounting for 50% and 39% of each group, respectively. Overall, 27.3% underwent splenectomy and 9% received embolization, with no statistical difference observed between the two groups. Elderly patients experienced lower rates of liver injury (28% vs. 37%, p-value 0.048) and less severe injuries (mean AAST score 2.21 vs. 2.65, p-value 0.018) with no difference in embolization or surgical management rates. There was no statistically significant difference in

**Table 2. Emergency department parameters univariable analysis.**

| Emergency department parameters | All patients (n = 1181) | Elderly group (n = 119) | Young group (n = 1062) | p- value |
|---|---|---|---|---|
| SBP(mmHg) +/- SD | 114.0 +/- 33 | 122.6 +/- 39 | 112.9 +/- 32 | **0.037** |
| DBP(mmHg) +/- SD | 68.4 +/- 24 | 71.2 +/- 23 | 68.0 +/- 24 | 0.396 |
| HR(/min) +/- SD | 96 +/- 25 | 90 +/- 25 | 96 +/- 25 | 0.093 |
| GCS | 13.72 | 13.72 | 13.18 | 0.959 |
| Hb(g/l) +/- SD | 126.0 +/- 26 | 118.9 +/- 26 | 126.7 +/- 25 | **0.004** |
| Arterial pH | 7.34 | 7.34 | 7.28 | **0.037** |
| Arterial lactates | 2.80 | 2.80 | 3.15 | 0.434 |

SBP: systolic blood pressure, DBP: diastolic blood pressure, HR: heart rate, GCS: Glasgow Coma Scale, Hb : haemglobin, SD : standard deviation), significant p-values in bold.

**Table 3. Abdominal organs injuries.**

| Abdominal organs injuries | All patients (n = 1181) | Elderly group (n = 119) | Young group (n = 1062) | p- value |
|---|---|---|---|---|
| | | Spleen | | |
| **Spleen injury n (%)** | 473 (40%) | 59 (50%) | 414 (39%) | **0.025** |
| **AAST score (mean)** | 2.93 | 2.64 | 2.97 | 0.126 |
| **Embolization n (%)** | 41 (9%) | 5 (8%) | 36 (9%) | 0.955 |
| **Splenectomy n (%)** | 129 (27%) | 12 (20%) | 117 (28%) | 0.201 |
| | | Liver | | |
| **Liver injury n (%)** | 425 (36%) | 33 (28%) | 392 (37%) | **0.048** |
| **AAST score (mean)** | 2.62 | 2.21 | 2.65 | **0.018** |
| **Embolization n (%)** | 13 (3%) | 2 (6%) | 11 (3%) | 0.606 |
| **Surgical management* n (%)** | 43 (10%) | 3 (9%) | 40 (10%) | 1 |
| | | Kidney | | |
| **Kidney injury n (%)** | 198 (17%) | 8 (7%) | 190 (18%) | 0.050 |
| **AAST score (mean)** | 2.68 | 1.5 | 2.72 | **0.001** |
| **Embolization n (%)** | 2 (1%) | 0 | 2 (1%) | 1 |
| **Surgical management* n (%)** | 14 (7%) | 0 | 14 (7%) | 0.926 |

N: number, AAST: American Association of the Surgery of Trauma, * Any surgical intervention (packing, hemostasis, resections, etc), significant p-values in bold.

the rate of kidney injuries and their management. However, younger patients had a higher AAST score (2.72 vs 1.50).(details on Table 3).

Regarding the outcomes, we reported a death rate of 13% in the EG and 7% in the YG (p-value 0.022). The requirement for operative treatment, blood transfusions and intensive care were similar in both group, as well as the length of stay in both intensive care and the hospital in general (Table 4).

Concerning the operative management, 68% of all the patients required surgery with little difference between both groups (YG: 68%, EG: 67%, p-value 0.8) and among the surgical patients, 62% in the YG and 55% in the EG had abdominal surgery (p-value 0.24). Of all surgical patients 11% had orthopedic, 2% thoracic and 1% neurological operations with no difference between the two groups.

Interestingly, after controlling for potential confounders by logistic regression, our multivariable analysis did not show that age was independently associated with increase in-hospital mortality (OR = 1.02, CI 95% = 0.98–1.05) or with emergency admission to the operating theatre (OR = 1, CI 95% = 0.99–1.02). Low GCS, the need of red blood cell transfusion, low arterial pH on admission, a high thoracic AIS (>3) and high ISS were associated with the need for operative management (Table 5).

**Table 4. Univariable analysis of outcomes.**

| Outcomes | All patients (n = 1181) | Elderly group (n = 119) | Young group (n = 1062) | p value |
|---|---|---|---|---|
| **OM (%)** | 806 (68) | 80 (67) | 726 (68) | 0.801 |
| **IC requirement** | 861 (73) | 91 (76) | 770 (73) | 0.356 |
| **IC LOS (+/-SD)** | 7.5 (+/-9.89) | 6.3 (+/- 7.9) | 6.7 (+/- 10.1) | 0.640 |
| **Hospital LOS (+/- SD)** | 15.4 (+/- 18.2) | 14.6 (+/-13.9) | 15.4 (+/- 18.6) | 0.631 |
| **BT requirement (%)** | 366 (31) | 43 (36) | 323 (30) | 0.705 |
| **Death (%)** | 95 (8) | 16 (13) | 79 (7) | **0.022** |

OM: operative management, IC: intensive care, LOS: length of stay, BT: blood transfusion, SD: standard deviation), significant p-values in bold.

**Table 5. Multivariable analysis for operative management.**

|  | OR | (95% CI) |
|---|---|---|
| **Age** | 1 | 0.99–1.02 |
| **Penetrating trauma** | 2.29 | 0.87–6.5 |
| **Shock** | 1.64 | 0.67–4.18 |
| **GCS** | **1.2** | **1.08–1.35** |
| **Hb** | 0.93 | 0.79–1.1 |
| **BT** | 0.54 | 0.33–1.07 |
| **pH** | **0.01** | **0–0.34** |
| **Arterial lactates** | 0.22 | 0.01–11.48 |
| **Abdominal AIS 3** | 0.79 | 0.35–1.73 |
| **Abdominal AIS 4–5** | 0.41 | 0.14–1.14 |
| **Thoracic AIS 3** | 0.49 | 0.21–1.15 |
| **Thoracic AIS 4–5** | **0.19** | **0.05–0.79** |
| **ISS** | **1.11** | **1.05–1.18** |

CI : confidence interval, GCS: Glasgow Coma Scale, Hb: haemoglobin, BT: blood transfusion, AIS: abbreviated injury score, ISS: injury severity score, OR: odd ratio).

Multivariable analysis for mortality showed that shock and need for red blood cell transfusion were associated with a higher mortality.

## Discussion

Our analysis concurs with existing literature that elderly patients suffering from abdominal trauma have a higher mortality for a similar mean abdominal AIS in comparison to younger patients (13% vs 7%). Interestingly, the results of the multivariable analysis indicate that age alone is not an independent factor that could explain the increased mortality.

Moreover, the mechanism of trauma in the elderly was different with seemingly less force implied, less traffic accidents but more falls. This supports the fact that older patients suffer more severe injuries with milder trauma.

However, the management seems to be similar in both group with a similar rate of operative management, blood transfusions and intensive care unit admissions.

We reported that elderly patients suffered from more severe thoracic trauma with a higher mean AIS. The existing literature reports that geriatric patients with thoracic trauma suffer from a higher mortality. For example a retrospective analysis of 808 patients with rib fractures showed a doubled mortality rate in the elderly in comparison to the adult population (18% vs 9%) [12]). In our study, the higher mortality in the elderly could be partially explained by thoracic injuries and their complications. Moreover, it is now well established that pulmonary complications are frequent after abdominal surgery. Age and emergency operations are independent risk factors of developing such postoperative complications [13].

Our analysis of the parameters on admission in the ED showed that older patients had higher systolic blood pressure and lower hb concentration, which could be explained by preexisting conditions such as hypertension and chronic anemia. As reported by James M. Bardes et al [14], there is an excessive undertriage in elderly patient suffering from trauma leading to an increased morbi-mortality. This could be explained by misleading normal vital signs, medications and poor physiologic reserve. The authors therefore concluded that older age should be considered as a criteria for trauma team activation even with a risk of over triage.

Logistic regression analysis did not show that age was independently associated with increased in-hospital mortality or emergency admission to the operating theatre, while shock and the need for red blood cell transfusion were associated with higher mortality.

Regarding these results, we cannot concluded that age itself is a predictor of mortality in abdominal trauma. It is unclear whether the higher mortality in the elderly patient group is due to preexisting comorbidities or due to normal physiological aging. For instance, the physiological response to stress is diminished in this population with an age-related decrease in cardiac index, maximum heart rate and adrenal function [15]. Krishenbom et al. published in 2017 a retrospective database analysis of 1027 patients aged 65 years and older suffering from any blunt trauma and concluded that a high ISS, preexisting medical conditions and medications are stronger predictors of morbidity than age itself [16].

The definition of geriatric patients with a threshold of 65 years is commonly accepted in trauma settings. The Advanced Trauma Life Support committee states that 25% of these patients appear to have preexisting conditions significantly impacting morbidity and mortality after trauma. However, a recent multicenter study of more than 250,000 trauma patients found a significant increase in mortality at three distinct threshold ages of 55, 77, and 82 years. The analysis showed that there was a statistically significant increase in comorbidities at the age of 55 [17]. The authors assessed that an age cutoff is not the most reliable marker but is a convenient substitute for a patient's overall medical condition.

Pre-existing comorbidities are associated with polymedication, which has been identified as a predictor of complications [18]. For example, oral anticoagulant medication is associated with a higher intra-abdominal organ injury rate and a higher 30-day mortality after trauma [19]. Other pharmaceutical agents such as antihypertensive drugs are associated with a higher risk of injury but also have an impact on the management of trauma patient [20].

There is an emerging recognition of the use of frailty, representing a diminished physiological reserve and response that is distinguished from aging and co-morbidity, to predict morbi-mortality in order to guide trauma patients' management. The use of assessment tool such as the Trauma-Specific Frailty Index and the Clinical Frailty Scale shows promising results in predicting outcomes [21, 22] but their use may not always be relevant in an emergency setting. A retrospective single-centre cohort review showed that older patients with high frailty score, suffering from major trauma (ISS>15), experienced a one year mortality of up to 51% and suffer from higher serious complication leading to a possible loss of autonomy [23].

## Limitations

It is important to note that this study has some limitations and potential sources of bias that may affect the results.

By its retrospective nature, our study is subject to information bias due to inaccurate data. We encountered a certain level of missing values, especially the vital signs and laboratory tests, which we considered missing completely at random and we performed a complete-case analysis to address them. Unfortunately, this assumption about missingness cannot be definitively confirmed and there is a possibility of bias.

Another source of bias could that the study only included patients from two tertiary care centers which may not be representative of the general trauma population.

Furthermore, the study did not control for other factors that could influence the outcomes such as patient's comorbidities which could have a significant impact on our results.

Regarding external validity, the sample size and the data collection from two hospitals, we acknowledge the generalizability of our results for patients in similar settings. However, our analysis is constrained by the lack of recent data, as the dataset utilized was collected until

2019, potentially limiting the applicability of the findings to current clinical practices and patient demographics.

## Conclusion

Our retrospective analysis showed an almost doubled mortality for similar abdominal injuries in older patients. However, older age was not an independent factor for higher mortality and the global management, including operative management and need of intensive care, were similar in both the younger and older group focusing our attention on a possible point of improvement. For instance, lowering vital signs threshold values for elderly patient and including frailty score criteria more than age by itself in abdominal trauma guidelines could improve outcomes of this population.

Surgeons and emergency department practitioners are expected to face an increasing number of elderly patients suffering from traumatic injuries. As shown in our study, these patients suffer from a higher mortality, which is to be expected in the setting of physiological changes, but in our opinion could explained by an undertriage and associated injuries especially of thoracic nature. Including frailty score in abdominal trauma guidelines after further studies should help reduce mortality and morbidity in the elderly allowing for better care for this particularly fragile group of patients.

## Author Contributions

**Conceptualization:** Jeremy Dash, Elisabeth Andereggen, Damien Massalou.

**Data curation:** Elisabeth Andereggen, Damien Massalou.

**Formal analysis:** Jeremy Dash, Imad Bentellis.

**Investigation:** Jeremy Dash.

**Methodology:** Jeremy Dash, Elisabeth Andereggen, Damien Massalou.

**Software:** Imad Bentellis.

**Writing – original draft:** Jeremy Dash.

**Writing – review & editing:** Jeremy Dash, Elisabeth Andereggen.

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
