## [Decision Letter · Decision Letter 0]

17 Jun 2024

PONE-D-24-11676Comparison of adult versus elderly patients with abdominal trauma: a retrospective database analysisPLOS ONE

Dear Dr. Dash,

Thank you for submitting your manuscript to PLOS ONE. After careful consideration, we feel that it has merit but does not fully meet PLOS ONE’s publication criteria as it currently stands. Therefore, we invite you to submit a revised version of the manuscript that addresses the points raised during the review process.

We look forward to receiving your revised manuscript.

Kind regards,

Yusuke Tsutsumi

Academic Editor

PLOS ONE

Journal Requirements:

Reviewers' comments:

Reviewer's Responses to Questions

**Comments to the Author**

1. Is the manuscript technically sound, and do the data support the conclusions?

Reviewer #1: Partly

Reviewer #2: Yes

2. Has the statistical analysis been performed appropriately and rigorously? 

Reviewer #1: Yes

Reviewer #2: Yes

3. Have the authors made all data underlying the findings in their manuscript fully available?

Reviewer #1: No

Reviewer #2: No

4. Is the manuscript presented in an intelligible fashion and written in standard English?

Reviewer #1: Yes

Reviewer #2: Yes

5. Review Comments to the Author

Reviewer #1: - Why is there no data available from 2019 till present date keeping in mind that the number of patients in the elderly group is quite low and there are only short-term outcome paramters analyzed?

- Data on the pattern of abdominal trauma is missing (e. g. how many splenic, hepatic or renal injuries appeared in the corresponding groups)

- Has there been interventional treatments in the analyzed patient collective e. g. coiling of splenic arteries which prevented the patient from surgical treatment?

- If data from certain parameters like vital signs are missing and patients were included in the study anyways, a number of patients analyzed for each group and paramter should be given (e. g. how many patients were analyzed regarding systolic blood pressure in the YG an EG).

Reviewer #2: Dear authors

Thank you for giving me the opportunity to review this interesting manuscript. This theme is very important. Your manuscript is well written. So I have only one comment.

Methods:

Not multivariate but multivariable analyses. It is the same for the after section.

“The multivariate analyses was adjusted to the following factors (age, penetrating injuries, shock, GCS, SBP, DBP, hb, arterial lactates, abdominal and thoracic AIS and ISS). We used the R statistical software to perform statistical analysis.”

Hidalgo B, Goodman M. Multivariate or Multivariable Regression? Am J Public Health. 2013;103(1):39-40. doi:10.2105/AJPH.2012.300897

6. PLOS authors have the option to publish the peer review history of their article (what does this mean?). If published, this will include your full peer review and any attached files.

Reviewer #1: No

Reviewer #2: No

---

## [Author Response · Author response to Decision Letter 0]

9 Jul 2024

Journal Requirements:

Response : 

• Continuous line numbers have been added.

• Page numbers appears now on the lower left hand corner.

• Figure citations have been corrected.

• Other manuscript body formatting guidelines have been double checked according to the modified April 2017 Manuscript body formatting guidelines document.

• Affiliation footnotes on title page has been modified according to the modified January 2017 Title,author, affiliations formatting guidelines document.

Response : 

We fully support the principles of transparency and data sharing in scientific research. However, the data used in our study includes sensitive information such as patient age, mechanism of trauma, year of the trauma, injuries and the length of stay in two recognizable hospitals. Due to the nature of these data points, there is a risk that individuals could be identifiable, thereby compromising patient confidentiality and privacy.

Our research adheres to stringent ethical guidelines and complies with the privacy regulations set forth by our institutions and the New Federal Act on Data Protection in Switzerland and the General Data Protection Regulation in France. These regulations impose strict limitations on the sharing of personal health information to protect patient privacy.

While we are unable to share the raw data publicly, we are committed to ensuring the transparency and reproducibility of our research. To this end, we can provide the contact information to which data requests may be sent :

For the Geneva University Hospital registry :

Dr Axel Gamulin 

Hôpitaux Universitaires de Genève 

Rue Gabrielle-Perret-Gentil 4

1211 Genève

axel.gamulin@hcuge.ch

0041223723311

Commission cantonale d'éthique de la recherche CCER 

Rue Adrien-Lachenal 8

1207 Genève

0041 22 546 51 01

ccer@etat.ge.ch

www.ge.ch/lc/ccer

For the Nice registry (REGISTRY CIL n°272) :

Dr Damien MASSALOU

Centre Hospitalier Universitaire de Nice

Voie Romaine 30

CS 51069 - 06001 Nice Cedex 1

massalou.d@chu-nice.fr

0033663269264 / 0033492038614

Non-author point of contact :

Commission nationale de l'informatique et des libertés

3 Place de Fontenoy

TSA 80715

75334 PARIS CEDEX 07

0033153732222

Review Comments to the Author

Reviewer #1 :

1. Why is there no data available from 2019 till present date keeping in mind that the number of patients in the elderly group is quite low and there are only short-term outcome paramters analyzed?

Response :

We acknowledge the concern regarding the data timeframe and the size of the elderly patient group. The data presented in our study were collected in 2020. At that time, we had access to patient records up until the end of 2019. Unfortunately, due to institutional and administrative constraints, we did not have access to more recent data beyond 2019 during our study period. We believe the revisions and responses provided have significantly improved the manuscript. We hope the reviewers find our responses satisfactory and that the manuscript is now suitable for publication.

We appreciate your interest in longer-term outcome parameters. The scope of our study was limited by the information available in our registries, which did not contain long-term follow-up data. Our registries are primarily designed for short-term outcome tracking, which includes immediate and early postoperative results. Unfortunately, they do not extend to longer-term outcomes.

We understand the importance of long-term data in providing a more comprehensive view of patient outcomes. As part of our future research initiatives, we plan to enhance our data collection methods to include longer-term follow-up information, provided we can secure the necessary resources and ethical approvals.

2. Data on the pattern of abdominal trauma is missing (e. g. how many splenic, hepatic or renal injuries appeared in the corresponding groups)

Response :

We have added the relevant data in the results section of the manuscript. The data regarding the number of patients suffering from splenic, hepatic and renal injuries and the occurrence of splenectomy and embolisation in both the elderly and younger groups, and the corresponding statistical analysis have been detailed.

3. -If data from certain parameters like vital signs are missing and patients were included in the study anyways, a number of patients analyzed for each group and paramter should be given (e. g. how many patients were analyzed regarding systolic blood pressure in the YG an EG).

We added in the method paragraph the number of patients with missing values.

Reviewer #2 :

1. Thank you for giving me the opportunity to review this interesting manuscript. This theme is very important. Your manuscript is well written. So I have only one comment.

Methods:

Not multivariate but multivariable analyses. It is the same for the after section.

“The multivariate analyses was adjusted to the following factors (age, penetrating injuries, shock, GCS, SBP, DBP, hb, arterial lactates, abdominal and thoracic AIS and ISS). We used the R statistical software to perform statistical analysis.”

Hidalgo B, Goodman M. Multivariate or Multivariable Regression? Am J Public Health. 2013;103(1):39-40. doi:10.2105/AJPH.2012.300897

Response :

We have corrected " -variate " to " -variable ". Thank you for your comment and for recommending the interesting article that explains the difference and highlights the confusion between the two terms.

---

## [Decision Letter · Decision Letter 1]

19 Jul 2024

PONE-D-24-11676R1Comparison of adult versus elderly patients with abdominal trauma: a retrospective database analysisPLOS ONE

Dear Dr. Dash,

Thank you for submitting your manuscript to PLOS ONE. After careful consideration, we feel that it has merit but does not fully meet PLOS ONE’s publication criteria as it currently stands. Therefore, we invite you to submit a revised version of the manuscript that addresses the points raised during the review process. Please submit your revised manuscript by Sep 02 2024 11:59PM. If you will need more time than this to complete your revisions, please reply to this message or contact the journal office at plosone@plos.org. Please include the following items when submitting your revised manuscript:A rebuttal letter that responds to each point raised by the academic editor and reviewer(s). You should upload this letter as a separate file labeled 'Response to Reviewers'.A marked-up copy of your manuscript that highlights changes made to the original version. You should upload this as a separate file labeled 'Revised Manuscript with Track Changes'.An unmarked version of your revised paper without tracked changes. You should upload this as a separate file labeled 'Manuscript'.If applicable, we recommend that you deposit your laboratory protocols in protocols.io to enhance the reproducibility of your results. Protocols.io assigns your protocol its own identifier (DOI) so that it can be cited independently in the future. For instructions see: https://journals.plos.org/plosone/s/submission-guidelines#loc-laboratory-protocols. Additionally, PLOS ONE offers an option for publishing peer-reviewed Lab Protocol articles, which describe protocols hosted on protocols.io. Read more information on sharing protocols at https://plos.org/protocols?utm_medium=editorial-email&utm_source=authorletters&utm_campaign=protocols.

We look forward to receiving your revised manuscript.

Kind regards,

Yusuke Tsutsumi

Academic Editor

PLOS ONE

Journal Requirements:

Additional Editor Comments:

I (the editor) have reviewed the revised manuscript for the points raised by Reviewer 1.

1. I found the lack of recent data was important problem, while I understand the restricted access to the data. Therefore, I strongly recommend to describe how this lack of up-to-date data affects the results in the Limitation section.

2.I think the manuscript have been much improved by including details of abdominal trauma in Table 3. However, the contents of Table 3 are inconsistent. For spleen injury, there is the information about the treatment (splenectomy, embolization). However, for the other two injuries, there is no information about treatments. For kidney injury, there is no information of AAST score. Please revise the contents.

3. Additionally, the layout of Table 3 is confusing, in which, injury site, AAST score and treatments are all arranged as same level of rows. Please revise the design of Table 3 using sub-headings.

4. As just a minor point, I think the last table is Table 5 not Table 4 (there are two Table 4s). Please revise.

Reviewers' comments:

Reviewer's Responses to Questions

**Comments to the Author**

1. If the authors have adequately addressed your comments raised in a previous round of review and you feel that this manuscript is now acceptable for publication, you may indicate that here to bypass the “Comments to the Author” section, enter your conflict of interest statement in the “Confidential to Editor” section, and submit your "Accept" recommendation.

Reviewer #2: All comments have been addressed

2. Is the manuscript technically sound, and do the data support the conclusions?

Reviewer #2: Yes

3. Has the statistical analysis been performed appropriately and rigorously? 

Reviewer #2: Yes

4. Have the authors made all data underlying the findings in their manuscript fully available?

Reviewer #2: Yes

5. Is the manuscript presented in an intelligible fashion and written in standard English?

Reviewer #2: Yes

6. Review Comments to the Author

Reviewer #2: I have no comment now.

7. PLOS authors have the option to publish the peer review history of their article (what does this mean?). If published, this will include your full peer review and any attached files.

Reviewer #2: No

---

## [Author Response · Author response to Decision Letter 1]

5 Aug 2024

Journal Requirements:

Response : 

All references have been reviewed. 

Reference number 8 (Sasser SM, Hunt RC, Faul M, Sugerman D, Pearson WS, Dulski T, Galli RL. Guidelines for field triage of injured patients: recommendations of the National Expert Panel on Field Triage, 2011. Morbidity and Mortality Weekly Report: Recommendations and Reports. 2012 Jan 13;61(1):1-20) : we noted that there is a more recent version reported by The American College of Surgeons. However there is no modification regarding the recommandation that patients over 65 with systolic blood pressure under 110 mmHg should be considered at high risk for serious injury and transported to the highest-level trauma centre.

The updated reference is : Newgard CD, Fischer PE, Gestring M, Michaels HN, Jurkovich GJ, Lerner EB, Fallat ME, Delbridge TR, Brown JB, Bulger EM. National guideline for the field triage of injured patients: recommendations of the National Expert Panel on Field Triage, 2021. Journal of Trauma and Acute Care Surgery. 2022 Aug 1;93(2):e49-60.

All other references have been checked and none has been retracted.

Additional Editor Comments: 

1. I found the lack of recent data was important problem, while I understand the restricted access to the data. Therefore, I strongly recommend to describe how this lack of up-to-date data affects the results in the Limitation section.

Response :

We acknowledge that the restricted access to more current data is an important issue, and we have discussed how this limitation may affect the results and their applicability to current clinical practices in the Limitations section.

2.I think the manuscript have been much improved by including details of abdominal trauma in Table 3. However, the contents of Table 3 are inconsistent. For spleen injury, there is the information about the treatment (splenectomy, embolization). However, for the other two injuries, there is no information about treatments. For kidney injury, there is no information of AAST score. Please revise the contents.

Response :

The content has been revised and we added the missing values and discussed the results. 

3. Additionally, the layout of Table 3 is confusing, in which, injury site, AAST score and treatments are all arranged as same level of rows. Please revise the design of Table 3 using sub-headings.

Response :

The design of Table 3 has been changed using sub-headings. Missing value have been added.

4. As just a minor point, I think the last table is Table 5 not Table 4 (there are two Table 4s). Please revise.

Response :

Correction has been made.

---

## [Editor Report · Decision Letter 2]

7 Aug 2024

Comparison of adult versus elderly patients with abdominal trauma: a retrospective database analysis

PONE-D-24-11676R2

Dear Dr. Dash,

We’re pleased to inform you that your manuscript has been judged scientifically suitable for publication and will be formally accepted for publication once it meets all outstanding technical requirements.

Kind regards,

Yusuke Tsutsumi

Academic Editor

PLOS ONE
---

## [Editor Report · Acceptance letter]

9 Aug 2024

PONE-D-24-11676R2 

PLOS ONE

Dear Dr. Dash, 

I'm pleased to inform you that your manuscript has been deemed suitable for publication in PLOS ONE. Congratulations! Your manuscript is now being handed over to our production team.

Kind regards, 

on behalf of

Dr. Yusuke Tsutsumi 

Academic Editor

PLOS ONE